# Structure and mechanism of a Type III CRISPR defence DNA nuclease activated by cyclic oligoadenylate

Stephen A. McMahon[1,3], Wenlong Zhu[1,3], Shirley Graham[1], Robert Rambo [2], Malcolm F. White [1]*  & Tracey M. Gloster [1]*

The CRISPR system provides adaptive immunity against mobile genetic elements in prokaryotes. On binding invading RNA species, Type III CRISPR systems generate cyclic oligoadenylate (cOA) signalling molecules, potentiating a powerful immune response by activating downstream effector proteins, leading to viral clearance, cell dormancy or death. Here we describe the structure and mechanism of a cOA-activated CRISPR defence DNA endonuclease, CRISPR ancillary nuclease 1 (Can1). Can1 has a unique monomeric structure with two CRISPR associated Rossman fold (CARF) domains and two DNA nuclease-like domains. The crystal structure of the enzyme has been captured in the activated state, with a cyclic tetra-adenylate ($cA_4$) molecule bound at the core of the protein. $cA_4$ binding reorganises the structure to license a metal-dependent DNA nuclease activity specific for nicking of supercoiled DNA. DNA nicking by Can1 is predicted to slow down viral replication kinetics by leading to the collapse of DNA replication forks.

[1] Biomedical Sciences Research Complex, School of Biology, University of St Andrews, North Haugh, St Andrews KY16 9ST, UK. [2] Diamond Light Source Ltd, Diamond House, Harwell Science and Innovation Campus, Fermi Ave, Didcot OX11 0DE, UK. [3] These authors contributed equally: Stephen A. McMahon, Wenlong Zhu. *email: mfw2@st-andrews.ac.uk; tmg@st-andrews.ac.uk

Type III (Csm/Cmr) CRISPR systems utilise a cyclase domain in the Cas10 subunit to generate cyclic oligoadenylate (cOA) by polymerising ATP[1–3]. The cyclase activity is activated by target RNA binding and switched off by subsequent RNA cleavage and dissociation[1,4]. cOA is a potent antiviral second messenger that sculpts the anti-viral response by binding to and activating CRISPR associated Rossman fold (CARF) family proteins[5]. The only CARF family effectors to be characterised to date are dimeric ribonucleases of the Csx1/Csm6 family, which have a C-terminal HEPN (Higher Eukaryotes and Prokaryotes, Nucleotide binding) domain that is allosterically activated by cOA binding in the CARF domain[6–9]. Csm6 from *Enterococcus italicus* and *Streptococcus thermophilus* are activated by cyclic hexa-adenylate (cA₆)[2,3], while Csx1 from *Sulfolobus solfataricus* and Csm6 from *Thermus thermophilus* are activated by the cyclic tetra-adenylate (cA₄) molecule[1,2,10]. In addition to ribonucleases, a wide range of CARF family proteins have been identified, including transcription factors such as Csa3[11], cA₄-degrading ring nucleases[12] and putative membrane proteins of unknown function[5,13], suggesting that much remains to be discovered. The cOA signalling pathway is crucial for defence against mobile genetic elements (viruses and plasmids) in multiple different systems[7,14–17]. In infected cells, activation of Csm6 results in degradation of both host and invading RNA, leading to significant reductions in cell growth until infection is cleared[17].

*T. thermophilus* has well characterised type III CRISPR systems and accessory proteins. The *T. thermophilus* Cmr (type III-B) complex was one of the first studied biochemically, revealing the distinctive cleavage of target RNA with 6 nucleotide spacing[18,19]. High resolution cryo-EM studies revealed the mechanistic basis of this RNA cleavage by the backbone subunit Cmr4[20]. The *T. thermophilus* Csm (Type III-A) complex was shown to possess both sequence specific RNA cleavage and RNA-dependent DNA degradation activities, fitting the emerging consensus for type III CRISPR defence[21]. The crystal structure of the *T. thermophilus* Csm6 (TTHB152) protein revealed a dimeric arrangement with a HEPN nuclease domain possessing weak ribonuclease activity and a CARF domain postulated to bind activating ligands[6]. This was subsequently confirmed with the discovery that the cyclase domain of Cas10 generates cyclic oligoadenylates that activate CARF domain proteins, including Csm6, which is stimulated by cA₄ binding to become a much more active ribonuclease[2]. A second Csm6 orthologue (TTHB144) was recently shown to be activated by cA₄ binding and also capable of cA₄ degradation—the first example of a self-limiting Csm6-family protein[10].

TTHB155, hereafter Can1 (CRISPR ancillary nuclease 1), was identified as a potential accessory protein of the type III CRISPR system by Shah *et al.* in 2018 (cluster 107)[22]. The gene is strongly up-regulated following phage infection in *T. thermophilus* and is situated adjacent to the CRISPR-5 locus, with operons encoding subunits of the Cmr and Csm complexes nearby[23]. Can1 is a large protein, 636 amino acids in length. Sequence analysis using HHPRED[24] suggested the presence of two CARF domains, one at the N-terminus and one towards the centre of the protein, together with a C-terminal PD-D/ExK family metal-dependent nuclease domain (Fig. 1a). The protein therefore has a number of unique features that have not been investigated previously. Here, we show that Can1 is a metal dependent DNA nuclease, activated by cA₄ binding. The structure of Can1 bound to cA₄ reveals extensive structural reorganisation around the activator, bringing the two nuclease-like domains together and forming a DNA binding interface with an active site for DNA cleavage. The enzyme preferentially nicks supercoiled DNA to generate ligatable products, which may slow down viral replication without causing catastrophic damage to the host genome.

## Results

**Overall structure of Can1 bound to cA₄.** A synthetic gene encoding Can1 was constructed and cloned into the *E. coli* expression vector pEV5HisTEV[25]. The protein was expressed with an N-terminal, TEV protease-cleavable polyhistidine tag, and purified by immobilised metal affinity chromatography and gel filtration, with concomitant tag removal, as described in the Methods. Given the observation that the *T. thermophilus* Csm6 ribonucleases are activated by cA₄[2,10], we reasoned that Can1 might bind the same activator. Can1, with selenomethionine incorporated, was crystallised in the presence of cA₄. Diffraction data were collected at Diamond Light Source to 1.83 Å resolution and phased using the incorporated selenium atoms. Following autobuilding of the model, it was refined to a final $R_{work}$ of 18% and $R_{free}$ of 21%. The protein chain is traced from residue 3 to 638, with three short sections too disordered to model (464-471, 491-494 and 530-539). Efforts were made to crystallise apo Can1, but no crystals were forthcoming.

The structure of Can1 is unique, with four distinct domains linked by flexible loops (Fig. 1). At the N-terminus there is a CARF domain (residues 3-151), domain 2 (residues 168-276), a second CARF domain (residues 293-422) and a nuclease domain (residues 426-638). The two CARF domains display a canonical Rossman fold, with the addition of a β-strand compared to other reported CARF domain structures; each domain comprises a core of five parallel β-strands alternating with four α-helices (Supplementary Fig. 1). This is followed by a further two β-strands that form a beta-hairpin, and an α-helix. The nuclease domain at the C-terminus comprises a central core of six β-strands flanked by six α-helices typical of the PD-D/ExK metal-dependent nuclease family[26]. Domain 2 comprises five β-strands flanked by four α-helices, reminiscent of part of the core domain of the PD-D/ExK nuclease family, with a helix-turn-helix motif between the second and third helices.

**cA₄ recognition by Can1.** There is unambiguous electron density in the $F_{obs}—F_{calc}$ map at 3σ corresponding to a molecule of cA₄ bound to Can1 (Fig. 2a), which was modelled following generation of the library in Acedrg[27]. The molecule of cA₄ sits at the interface between the two CARF domains of Can1 and is completely enclosed by the protein (Fig. 1c). There are a fairly modest number of hydrogen bond interactions between Can1 and cA₄ given its size, all but one of which originate from the two CARF domains (Fig. 2b and Supplementary Fig. 2). For clarity, the AMP moieties in cA₄ have been arbitrarily named A1-4. There are hydrogen bonds formed between three of the AMP moieties in cA₄ (A1, A2 and A3) and residues in the first CARF domain; these interactions are formed via main chain atoms of residues Lys90, Asn12, Asp13 and Gly88, and side chain atoms of residues Lys90, Asn12, Tyr143, His113 and Tyr147. Likewise, there are hydrogen bond interactions formed between three of the AMP moieties in cA₄ (A1, A3 and A4) and residues in the second CARF domain; these are formed via main chain atoms of residues Thr384, Gln301, Gly405 and Asn380, and side chain atoms of residues Tyr402, Thr384, Glu324, Thr322, Ser299 and Asn380. A total of 19 water molecules also hydrogen bond directly with cA₄.

Superposition of the four adenosine units (centred on the carbon atoms in the ribose sugar) within cA₄ show a marked difference in the conformation of one of them (Supplementary Fig. 3). Whilst three of the adenosine moieties superimpose well (with respect to the ribose sugar and imidazole moiety of the adenine base; one adenine base is flipped compared to the other two but in the same plane), the adenine of the other adenosine moiety (A1) is in a more axial position at the anomeric carbon. This adenine base displays a clear π-π stacking interaction with

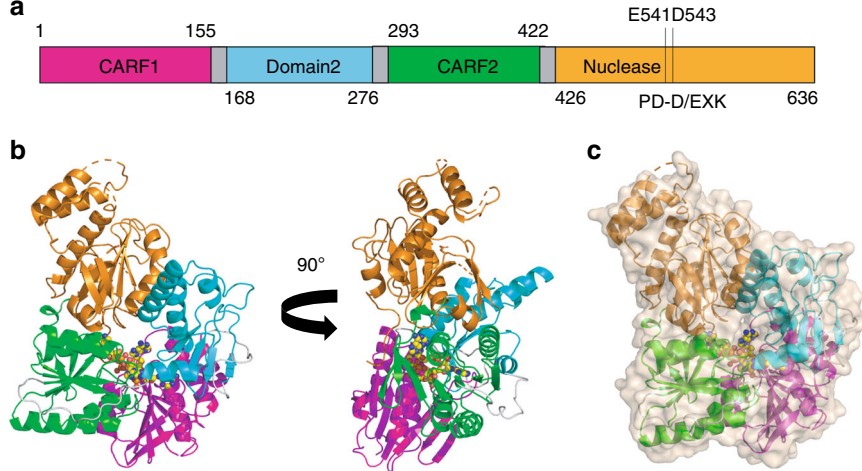

**Fig. 1 Domain organisation and structure of Can1. a** Domain organisation of Can1 with residue numbers indicated. Residues Glu541 and Asp543 are in the nuclease active site. **b** Two views of the overall structure of Can1 shown in cartoon representation. Coloured as in panel (**a**): first CARF domain (magenta), domain 2/nuclease-like domain (cyan), second CARF domain (green), nuclease domain (orange), loops between domains (grey). A molecule of $cA_4$ bound to Can1 is shown in sphere representation (carbon in yellow, nitrogen in blue, oxygen in red). **c** Overall structure of Can1, as coloured in panel (**b**), with surface representation to illustrate $cA_4$ is completely buried.

Trp42, which appears to be the driving force for this alternate, and most probably higher energy, conformation for the adenine. It has the knock-on effect of causing the ribose sugar to display a 2'-*exo* conformation, compared to an 2'-*endo* conformation which is observed in the other adenosine units.

Only two residues from the nuclease domain and domain 2 are in close proximity to $cA_4$. Gly550, from the nuclease domain, forms a hydrogen bond interaction via its main chain carboxyl group to the adenine moiety of A4 (Fig. 2b). Gln222 from domain 2 does not interact directly with any atoms in $cA_4$, but hovers above the centre of the $cA_4$ molecule, and makes two water-mediated interactions to opposite phosphate groups in $cA_4$ (Supplementary Fig. 4). To investigate the importance of these interactions for ligand binding, we engineered and purified the CARF domain variants N12A, W42A, K90E, H113A and Q222E. The effects of these changes on $cA_4$ binding were monitored by electrophoretic mobility shift assay (EMSA) using labelled $cA_4$. Binding of $cA_4$ was abolished for the K90E variant, severely reduced for the N12A variant, and moderately affected for W42A, H113A and Q222E (Supplementary Fig. 5). These observations highlight the distributed nature of $cA_4$ interactions and the crucial role of Lys90, which is situated at the base of the $cA_4$ binding pocket.

All other characterised CARF family proteins consist of a single CARF domain, which homo-dimerises to bind one molecule of cOA. Can1 is unique as it possesses two CARF domains in a single polypeptide. The CARF domains superimpose with an RMSD of 2.5 Å over 113 Cα atoms, although few residues are conserved (Supplementary Figs. 6 and 7A). A striking feature of $cA_4$ recognition is the asymmetry of the interactions formed with Can1, as the interactions with each AMP moiety are distinct. The comparison highlights the structural equivalence of Asp13 and Gln301, which both make main chain interactions with $cA_4$ (A1 and A3, respectively), and of Lys90 and Thr384 (A2 and A4, respectively), which both make main chain and side chain interactions with $cA_4$.

**Structural comparisons of the $cA_4$ binding site.** DALI[28] searches for the nearest structural homologue to each CARF domain in Can1 gave identical results for each: hypothetical protein VC1899 (PDB: 1XMX), a CARF family protein that is a member

of the DUF1887 family. The CARF dimer (taken as the two domains from a single polypeptide chain) of Can1 superimposes with the CARF dimer (formed by two monomers) of VC1899 with an RMSD of 3.6 Å over 294 Cα atoms (Fig. 3a). This superimposition allows exploration of likely interactions between VC1899 and $cA_4$. All of the main chain interactions observed between Can1 and $cA_4$ are likely conserved, as are a number of the side chain interactions (Supplementary Figs. 6 and 7B). There is no equivalent in VC1899 to Trp42, the residue responsible for causing one of the adenine bases (from A1) to sit in a more axial conformation with respect to the ribose sugar. With the absence of this interaction this adenine may not be in such a strained position when bound to VC1899, and there is sufficient room in the binding site for this to fit in a more relaxed position (Fig. 3b). Few close interactions/clashes between $cA_4$ and VC1899 are predicted, and these involve side chains which could easily be repositioned, suggesting there is little rearrangement of either the CARF dimer or of residues in the CARF domains upon binding of $cA_4$. In other words, with the caveat that VC1899 has not yet been demonstrated to bind $cA_4$, the $cA_4$ binding site of Can1 is likely to be largely pre-formed in the apo-protein.

**The nuclease and nuclease-like domains.** DALI searches for structural homologues of the nuclease domain of Can1 once again gave the hypothetical protein VC1899 (PBD: 1XMX) as the highest hit. The nuclease domain from Can1 superimposes with the nuclease domain of VC1899 with an RMSD of 3.4 Å over 160 Cα atoms (Fig. 3c, with nuclease active site residues shown). Whilst the core of the nuclease domain is very similar, Can1 has an extra helix-loop-helix feature between residues 441 and 498, whereas at the equivalent position (residues 156-248) VC1899 possesses a non-superimposable section. The active site residues Glu541, Asp543, Glu560 and Lys562 in Can1 overlap with equivalent active site residues in VC1899. The closest structural homologues with dsDNA bound are the type 2 restriction enzymes *Age*I (PDB: 5DWB[29]) and *Ngo*MIV (PDB: 1FIU[30]), which overlap with the nuclease domain of Can1 with an RMSD of 4.0 Å over 143 Cα atoms and 3.4 Å over 133 Cα atoms, respectively (Supplementary Fig. 8). The core secondary structure elements of the nuclease domain of Can1 and each of the restriction enzymes are conserved, and the active site residue

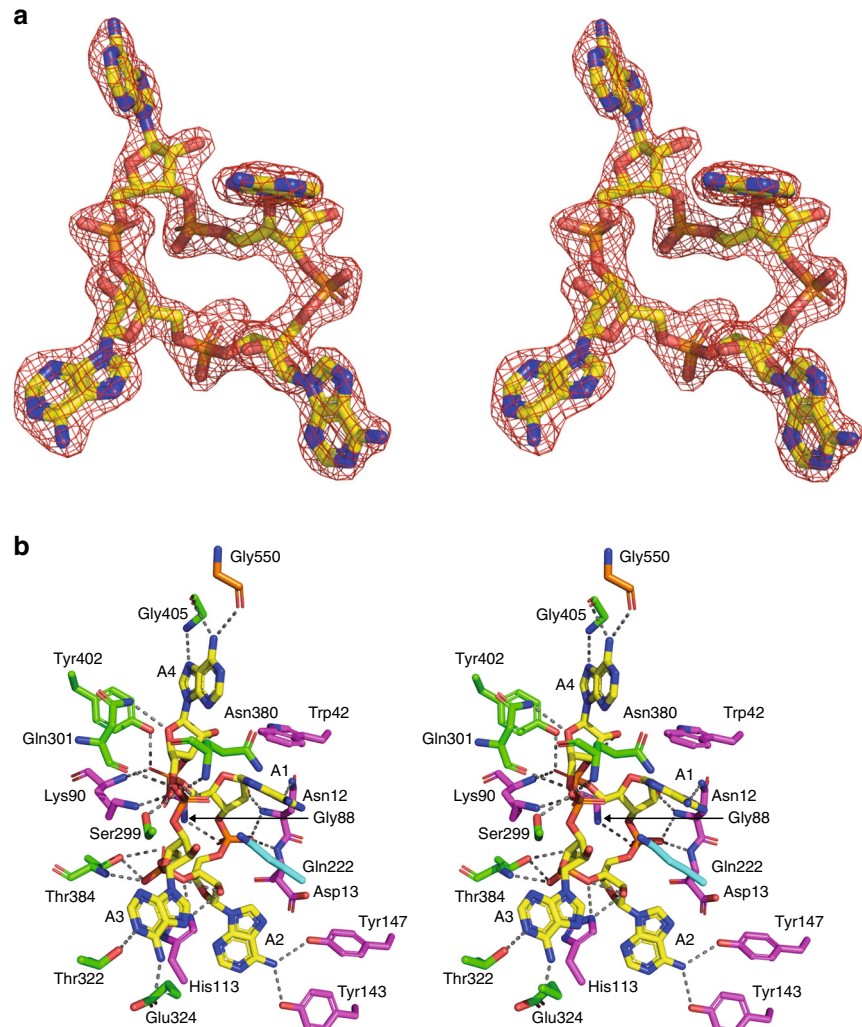

**Fig. 2 Structure of the cA₄ activator and interactions with Can1. a** Divergent stereo representation of cA₄ (in stick representation; carbon in yellow, nitrogen in blue, oxygen in red) with the maximum likelihood/$\sigma_A$ weighted $F_{obs}-F_{calc}$ electron density map contoured at 3.0 sigma shown in red. **b** Divergent stereo representation of the binding site of cA₄ in complex with Can1. cA₄ (with each AMP numbered A1-A4) and binding site residues are shown in stick representation, with cA₄ in yellow and amino acid residues coloured by the domain from which they originate (CARF domain 1: magenta, domain 2/nuclease-like domain: cyan, CARF domain 2: green, nuclease domain: orange). The dotted lines represent hydrogen bond interactions.

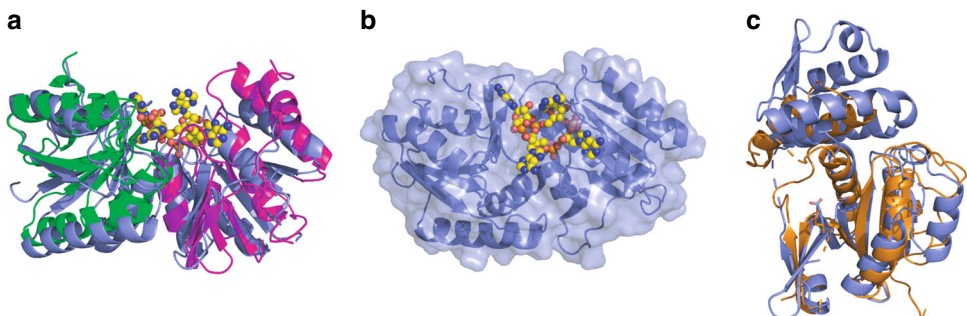

**Fig. 3 Structural comparison of Can1 and the DUF1887 family protein VC1899. a** Superimposition of the CARF domains from Can1 (coloured as in Fig. 1) in complex with cA₄ (yellow spheres) with the CARF domain from VC1899 (mauve). The VC1899 dimer (monomer plus the symmetry mate that forms the functional dimer) was overlapped with Can1. **b** Surface and cartoon representation of VC1899, with cA₄ (yellow spheres) modelled based on the superimposition shown in panel (**a**). **c** Superimposition of the nuclease domain from Can1 (orange) and the nuclease domain from VC1899 (mauve). The active site residues are shown as sticks.

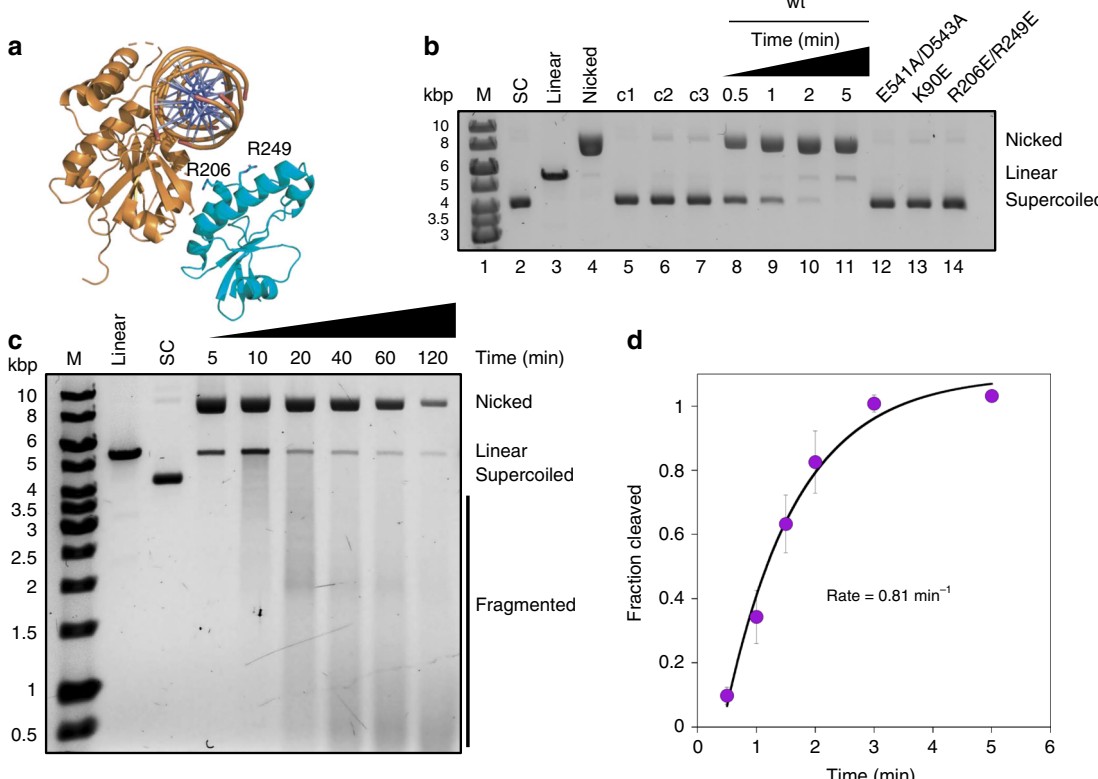

**Fig. 4 Can1 nicks supercoiled plasmid when activated by cA4 in the presence of MnCl2. a** The nuclease domain of Can1 (orange) in complex with the modelled position of dsDNA (based on homologous structures, the nuclease domains of *Age*I (PDB: 5DWB) and *Ngo*MIV (PDB: 1FIU)). The nuclease-like domain of Can1 is shown in cyan and Arg206 and Arg249 are shown as sticks. **b** Native agarose gel analysis of plasmid (2 nM) cleavage by Can1 (200 nM), showing rapid nicking in the presence of cA4 (200 nM) and MnCl2 (5 mM) (lanes 8-11). Standards corresponding to supercoiled, linear and partially nicked plasmid are shown in lanes 2-4. Lanes 5-7 show the reactions incubated for 5 min without protein (c1), MnCl2 (c2) or cA4 (c3), respectively. Can1 variants E541A/D543A, K90E and R206E/R249E generate minimal nicked product after 5 min (lanes 12-14). M—DNA size markers. **c** Extended time course of plasmid degradation by Can1. Rapid nicking is followed by slower degradation to small linear products over 2 h. Standards are as described in part **b**. **d** Single-turnover kinetic analysis of supercoiled plasmid nicking by Can1, under the conditions in **b**, yield a rate constant of $0.81 \pm 0.15$ min$^{-1}$. Data points are the mean of triplicate experiments with the standard deviation shown. Source data are provided as a 'Source Data' file.

Asp543 in Can1 overlaps with Asp142 in the *Age*I nuclease and Asp140 in the *Ngo*MIV nuclease. Thus, the approximate binding position of the dsDNA in complex with Can1 can be predicted with some confidence (Fig. 4a).

Perhaps more surprisingly, DALI searches revealed that the top hit for homologues of domain 2 of Can1 is also VC1899, which overlaps with an RMSD of 2.5 Å over 86 Cα atoms. Domain 2 of Can1 also superimposes with the nuclease domain of Can1 with an RMSD of 2.6 Å over Cα 83 atoms, but it lacks the canonical nuclease active site residues of either (Supplementary Fig. 9). We herewith refer to domain 2 as the "nuclease-like domain".

**Can1 is a cA4 activated, metal dependent DNA nuclease**. The variants E541A/D543A, targeting the PD-D/ExK active site, K90E, targeting a key interaction with cA4 in the CARF domain, and R206E/R249E, targeting a putative DNA binding site in the nuclease-like domain, were expressed and purified as for the wild-type enzyme. Wild-type Can1 (200 nM) cleaved a supercoiled plasmid substrate (2 nM) in the presence of 200 nM cA4 and 5 mM MnCl2 (Fig. 4b). The majority of the plasmid was nicked within 1 min, with a small amount linearised after 5 min incubation. Reactions in the absence of MnCl2 or cA4 showed very little activity, confirming that Can1 is a metal dependent DNA nuclease activated by cA4. Moreover, variants E541A/D543A (nuclease variant) and K90E (CARF domain variant) were both

catalytically inactive, as expected for a cA4-activated PD-D/ExK family nuclease. The R206E/R249E variant, which reverses two positive charges on the surface of the nuclease-like domain, was also inactive, suggesting a role for this domain in DNA binding. These residues are in reasonably close proximity to the predicted position of the dsDNA (Fig. 4a), and the nuclease-like domain may move closer still upon DNA binding. Moderate reductions in cA4 cleavage were observed for the other cA4-interacting residue variants tested (Supplementary Fig. 5B).

On extended incubation over 2 h, Can1 slowly degraded the nicked plasmid to linear fragments of progressively smaller size (Fig. 4c). Thus, the enzyme favours nicking of supercoiled DNA, with more extensive DNA degradation seen only under extended incubation. To measure the rates of supercoiled plasmid nicking under pseudo-single turnover conditions, 200 nM protein was incubated with 2 nM plasmid, and the fraction of supercoiled plasmid cleaved was quantified, normalised, plotted and fitted to an exponential equation as described in the Methods. Two biological replicates and six technical replicates were carried out, yielding a rate of $0.81 \pm 0.15$ min$^{-1}$ (Fig. 4d).

**Can1 cleaves supercoiled DNA at random sites**. To determine whether Can1 nicks supercoiled DNA at a specific site, we compared its activity to the Nickase Nt.*Bsp*QI, which nicks DNA specifically at 5'-GCTCTTCN/ sites, and has one recognition site

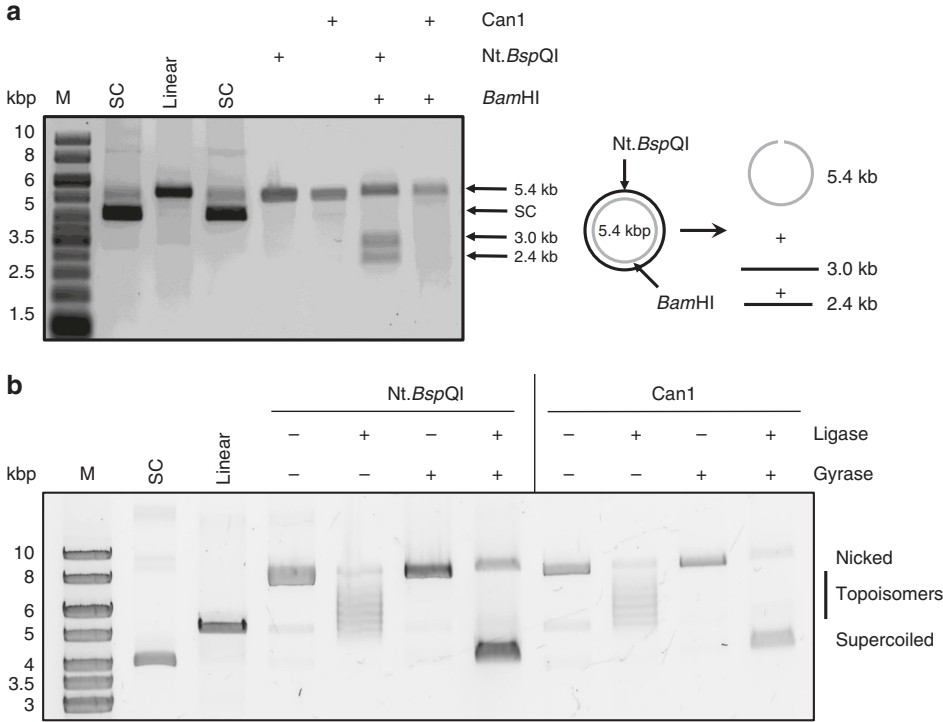

**Fig. 5 Can1 nicks plasmid DNA at random sites, leaving ligatable DNA ends. a** Plasmid nicking by the Nickase Nt.*Bsp*QI and Can1 were analysed by alkaline agarose gel electrophoresis where all DNA is denatured into single-stranded products. Both generate linear 5.5 kb products. When Nt.*Bsp*QI nicking at its recognition site is followed by *Bam*HI digestion, the expected linear products of 3.0 and 2.5 kb are also observed. Can1, which nicks at random sites, generates a wide range of linear products following *Bam*HI digestion that appear as a smear in the gel. **b** Both the Nickase Nt.*Bsp*QI and Can1 nick supercoiled plasmid to yield open circle DNA that can be visualised following agarose gel electrophoresis. Ligation of these products by DNA ligase generates closed circular DNA with a range of topoisomers. Addition of DNA gyrase restores the supercoiled DNA substrate. Control lanes are as for Fig. 4. Source data are provided as a 'Source Data' file.

in the plasmid pEV5HisTEV. By using alkaline agarose gel electrophoresis to analyse the cleavage products, we ensured that DNA migrated in single stranded form. Nt.*Bsp*QI nicking generated a single product band of 5.4 kb, as expected (Fig. 5a), whilst cleavage by Nt.*Bsp*QI followed by *Bam*HI (which also has a single recognition site in pEV5HisTEV) yielded 3 bands—a 5.4 kb band representing the strand that is cut only by *Bam*HI, and two smaller bands of 3.0 and 2.4 kb that result from cleavage of the Nt.*Bsp*QI nicked strand by *Bam*HI, as shown in the schematic. Can1 also generated a nicked strand of 5.4 kb, but when this product was further digested by *Bam*HI a smear of smaller DNA products were observed rather than specific fragments (Fig. 5a). This demonstrates that Can1 nicks supercoiled DNA at random sites.

Nucleases within the superfamily which contains the PD-D/ExK motif typically catalyse phosphodiester bond cleavage by a two metal ion catalytic mechanism[26]. A water molecule is deprotonated by one metal ion and used as a nucleophile to attack the scissile phosphate bond, generating 5'-phosphate and 3'-hydroxyl products[26]. To determine whether Can1 conformed to this mechanism, we incubated the nicked plasmid generated by Can1 with DNA ligase, followed by DNA gyrase, as described in the Methods. Commercial nicking endonuclease Nt.*Bsp*QI was used as a positive control. After incubating nicked plasmid with ligase, various relaxed topoisomers were observed (Fig. 5b), suggesting that the nicked plasmid was ligated to covalently-closed circular plasmid. The products were further incubated with DNA gyrase to introduce negative supercoils (Fig. 5b), confirming that ligation of the nicked sites had occurred. Thus, Can1 hydrolyses phosphodiester bonds in supercoiled DNA substrates, generating ligatable 5'-phosphate and 3'-hydroxyl products.

**Structural rearrangements upon cA4 binding**. The binding of a ligand as large as cA4 will inevitably require structural rearrangements of Can1, particularly as it is completely buried in the complex. As we lacked a crystal structure of apo Can1, we investigated the solution state of apo Can1 and Can1 bound to cA4 using small angle X-ray scattering (SAXS). Size exclusion chromatography coupled SAXS was performed on apo Can1 to ensure optimal background subtraction and sample characterisation, and a single, well defined elution peak was observed. The SAXS derived radius-of-gyration, $R_g$, shows the apo state is less compact than Can1 in complex with cA4, as characterised by a larger $R_g$ ($28.44 \pm 0.13$ Å vs $30.13 \pm 0.15$ Å), particle volume and maximum dimension, $d_{max}$. The dimensionless Kratky plot (Fig. 6a) suggests Can1 is not globular in either the bound or apo state, which is consistent with the observations of the X-ray crystal structure of Can1 bound to cA4. Furthermore, analysis of the real-space pairwise distances within the particles, P(r)-distribution, shows the apo state of Can1 has a wider overall distribution (Fig. 6b) with a substantially larger maximum dimension, $d_{max}$ (121.5 Å vs 101.5 Å for the complex), suggesting there is a significant conformational difference between the bound and apo states in solution. In fact, fitting the Can1 crystal structure to the SAXS data results in a Chi$^2 > 80$.

To investigate the structural changes required to explain the SAXS dataset of the apo state, we performed a distance restrained, rigid body molecular dynamics simulation that restricted the search to the distances within the apo state P(r)-distribution. The Can1 crystal structure demonstrated clearly defined domains that were used to define rigid bodies to which centre-of-mass restraints were assigned. Given the observed conservation of the CARF dimer conformation in the cA4 bound Can1 and apo

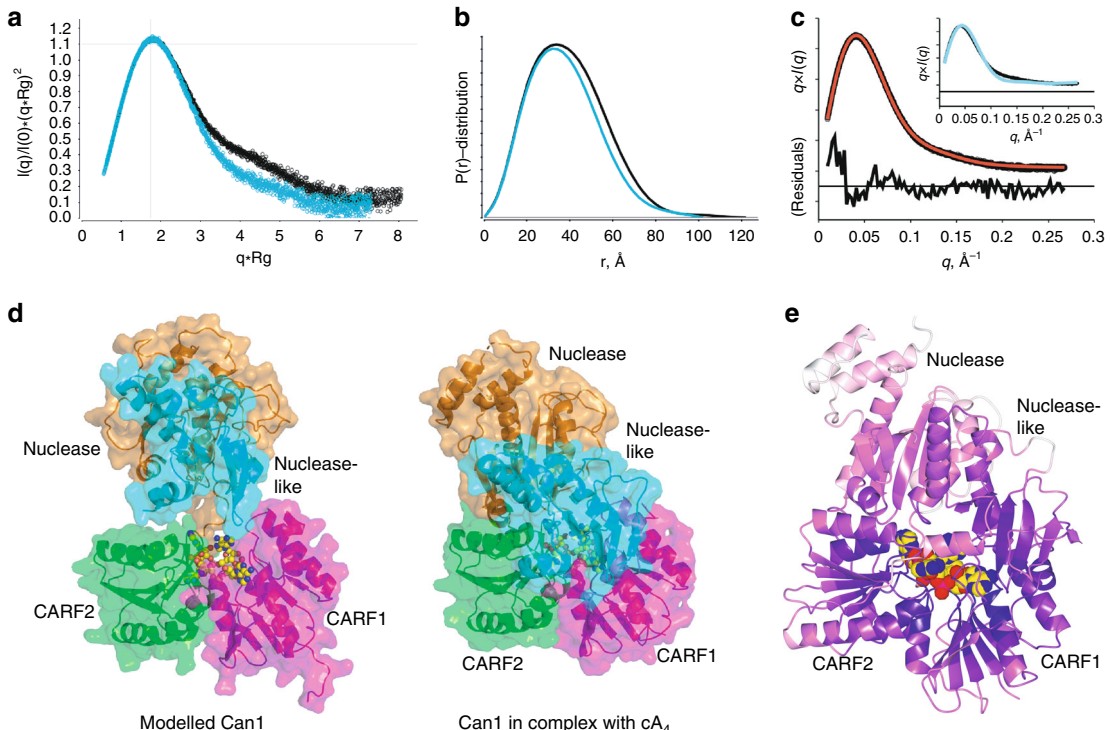

**Fig. 6 Structural rearrangements of Can1 upon cA$_4$ binding in solution. a** Dimensionless Kratky plot of Can1 in the presence (blue) and absence (black) of cA$_4$. Data were normalised with R$_g$ and I(0) of 28.88 ± 0.13 Å and 1.090E-1 ± 1E-4 respectively for Can1 with cA$_4$ and 30.26 ± 0.11 Å and 8.99E-2 ± 6.5E-05 respectively for apo Can1. Cross-hairs mark the Guinier-Krakty point or peak position for an ideal, globular particle. **b** Paired-distance, P(r), -distribution function performed as an indirect Fourier transform using the Moore method. P(r)-distributions were normalised to particle's Porod volume using 121,421 Å$^3$ and 132,291 Å$^3$ for the Can1 with cA$_4$ (blue) and apo Can1 (black), respectively. Widening of the distribution for apo Can1 suggests larger distances are more accessible. **c** Fit of apo Can1 SAXS data (black circles) using atomistic model from constrained torsion angle molecular dynamics. Best single-state model (red line), χ$^2$ 1.8, identifies a model that opens through translation of the nuclease domain. The best open model was achieved where the CARF dimer was maintained as a rigid body and the nuclease-like domain was allowed to vary ± 5 Å from its observed position. The nuclease domain was treated as a separate, rigid body tethered by an unconstrained loop (residues 423 to 428). Residuals of the fit are presented. The inset shows the fit of the Can1 crystal structure (cyan) to apo-state SAXS data (black). **d** Surface representation of Can1 as predicted by SAXS in the absence of cA$_4$ (left) (cA$_4$ is shown to illustrate the binding position and demonstrates the CARF domains are immobile) and the structure of Can1 in complex with cA$_4$ solved by X-ray crystallography (right). Coloured as first CARF domain (magenta), nuclease-like domain (cyan), second CARF domain (green), nuclease domain (orange). Loops between the domains are not shown for clarity. A molecule of cA$_4$ bound to Can1 is shown in yellow spheres. **e** The crystal structure of Can1 shown in cartoon form coloured by temperature factor from white (high temperature factor) to dark purple (low temperature factor). A molecule of cA$_4$ bound to Can1 is shown in yellow spheres.

VC1899 structures, we treated this unit as a rigid body. We generated ~2500 conformations that were subsequently fit to the apo state SAXS data. A best single-state model was found (best χ$^2$ 1.81; Fig. 6c, d), where there is movement of the nuclease and nuclease-like domains away from the core CARF dimer. This has the effect of exposing the cA$_4$ binding pocket to solvent, which is required for binding. Analysis of the temperature factor of residues in Can1 (Fig. 6e) shows the nuclease domain residues have higher average temperature factors than the other domains in Can1, which supports the hypothesis that the nuclease domain is more mobile. We envisage Can1 in solution is sampling between the open and closed states, exposing the cA$_4$ binding site in the CARF dimer. Upon cA$_4$ binding, the closed conformation of the protein is stabilised, as observed in the crystal structure. A potential 'hinge' between the second CARF domain and the nuclease domain to allow these open and closed states is the connecting loop which contains a Gly423/Arg424/Pro425 motif. The loop between the first CARF domain and the nuclease-like domain is longer, and contains two 'PPG' motifs (Pro160/Pro161/Gly162 and Pro165/Pro166/Gly167) which are likely to be important in providing flexibility between the domains. A superimposition of Can1 onto VC1899, structurally aligned on the CARF domain dimer from each protein only, suggests the possibility of movement of the nuclease and nuclease-like domains between the open and closed states (Supplementary Fig. 10). It is possible that a complex population of open and closed structures exists in the absence of cA$_4$.

## Discussion

The structure of the Can1 enzyme clearly demonstrates that the two halves of the protein, comprising a CARF domain N-terminal to a nuclease fold domain, are related to one another. Coupled to the observation that each half is clearly related to the DUF1887 family, as exemplified by the VC1889 protein, we can postulate a scenario whereby gene duplication, fusion and subsequent divergent evolution of an ancestral DUF1887 family member gave rise to the Can1 family. Can1 appears to be limited to members of the genus *Thermus*, but in contrast DUF1887 is much more widely distributed, with representatives found throughout the bacterial phyla and is particularly common in the proteobacteria. Although no member of the DUF1887 family has yet been characterised biochemically, given the relationship with Can1 and the conservation of a nuclease active site we suggest the provisional name "Can2" (CRISPR ancillary nuclease 2) for this family.

Analysis of the structure of a CARF domain protein bound to its cognate cyclic oligoadenylate ligand can be used to draw some general conclusions about the mechanism of recognition. Can1 is unusual in being a monomeric protein with two non-identical CARF domains, and therefore possesses less intrinsic symmetry than most family members. $cA_4$ is bound in a markedly asymmetrical conformation, which is in part exaggerated by the position of one adenine base in an axial position, driven by a stacking interaction with a tryptophan residue. Many different residues, the vast majority from the two CARF domains, contribute to the ligand binding pocket, which appears to be largely pre-formed in the apo protein. A significant number of these interactions are with protein main chain atoms, and although some of these are structurally conserved between the two 'halves' of Can1, the difference in side chain residue meant they would not have been predicted by sequence alone. Site-directed mutagenesis has highlighted a key role of Lys90, which occupies a central position in CARF domain 1 at the N-terminus of an α-helix, by forming a salt bridge with a bridging phosphate of $cA_4$; basic residues at an equivalent position have been shown to be essential for $cA_4$ recognition in other CARF family proteins[10,12].

Recently, the structures of two CARF-HEPN family ribonucleases, Csm6 from *Thermococcus onnurineus* and Csx1 from *S. islandicus*, have been solved with $cA_4$ bound[31,32]. Although each of these proteins have the basic unit of a CARF domain for $cA_4$ binding in common with Can1, there is little sequence conservation across the three proteins. In particular, Can1 lacks conserved Y14, Y19, S51, H155, and N158 $cA_4$-interacting residues identified as conserved across the Csx1/Csm6 family[32]. N158 is replaced by K90 in Can1, reflecting the importance of this position in the centre of the CARF domain for $cA_4$ binding despite a lack of sequence conservation. Residues W14 and H132, identified as important for $cA_4$ cleavage by Csm6[31], are also not conserved in Can1, and we see no evidence for ring nuclease activity by Can1 in vitro. The three enzymes bind $cA_4$ in different orientations, emphasising the conformational flexibility possible for this large cyclic nucleotide.

The $cA_4$ ligand is completely buried in the Can1 complex, demonstrating that significant protein conformational changes are required to facilitate binding. The close structural agreement observed for the two CARF domains of Can1 compared to the VC1899 apo protein suggests that the CARF dimer conformation does not change radically upon $cA_4$ binding. In contrast, the temperature factors of the residues in the crystal structure, coupled with SAXS analysis, suggest that the nuclease domain is mobile, which would help facilitate $cA_4$ binding to apo Can1. Modelling of dsDNA binding by the nuclease domain, coupled with the observation that conserved basic residues in the nuclease-like domain are essential for activity (presumably due to a role in DNA binding rather than directly in catalysis), suggests that $cA_4$ binding allows the nuclease and nuclease-like domains to assemble into a functional dsDNA binding site, activating the enzyme. Modelling of dsDNA from homologous structures suggests a further modest closure of the cleft between the two domains may occur upon DNA binding, although this will depend on the precise position of the dsDNA substrate bound to Can1.

An emerging theme of CRISPR-based defence systems is the role of collateral (non-specific) nucleic acid degradation in immunity (reviewed in ref. [33]). In addition to the cOA activated, non-specific ribonucleases described above for type III systems, type VI (Cas13) effectors provide effective immunity against mobile genetic elements by degrading RNA non-specifically[34,35]. The mechanism for this immunity was recently shown to result in degradation of host, rather than virus, RNA, driving infected cells into dormancy and thus preventing the spread of infection[36].

DNA is also a target for collateral degradation. Type III effectors hydrolyse ssDNA on binding target RNA—an activity that appears largely non-specific in vitro[37–39], but which may favour transcribing phage DNA in vivo[40]. Likewise, type V effectors such as Cas12a possess a non-specific ssDNA degradation activity that is licensed by crRNA-dependent DNA binding[41,42].

Here, we have described another option available to CRISPR defence systems: cyclic oligoadenylate-dependent supercoiled DNA cleavage by the Can1 enzyme. Once activated by $cA_4$ generated in response to detection of foreign RNA, Can1 cleaves supercoiled DNA to generate nicked products. In vivo, such an activity is likely to slow down the replication kinetics of rapidly-replicating phage, where nicks can result in the collapse of DNA replication forks to generate double-strand breaks. In contrast, such nicks are not too great a burden for the slowly-replicating host chromosome as they are easily repaired by DNA ligase. This nickase activity will operate in parallel with non-specific RNA degradation by the *T. thermophilus* Csm6 enzymes (Fig. 7), providing layered defence against mobile genetic elements detected by the CRISPR system. Given the structural conservation of CARF and nuclease domains in the Can2 (DUF1887) family, we predict that this family, which is widespread in bacteria, functions in a similar way to Can1 by cleaving supercoiled DNA. However, there is a key difference—the Can2 homodimer has two nuclease active sites, which may have functional consequences. In conclusion, we have revealed key molecular details of the cOA-mediated anti-viral defence mediated by type III CRISPR systems —one of the most common types of CRISPR effectors found in nature.

## Methods

**Expression and purification of Can1 and variants**. Two overlapping gBlock fragments to form the synthetic *can1* gene were purchased from Integrated DNA Technologies, Coralville, IA, United States (IDT) and cloned into the pEV5HisTEV vector using *Nco*I and *Hind*III restriction enzyme sites[25]. The cloning strategy resulted in the addition of two residues (Leu and Glu) to the C-terminus of the protein. The full sequence of the synthetic gene is shown in Supplementary Table 1. The variants E541A/D543A, K90E, N12A, W42A, H113A, Q222E, and R206E/R249E were generated using the QuikChange Site-Directed Mutagenesis kit as per manufacturer's instructions (Agilent Technologies). Primer sequences for mutagenesis are shown in Supplementary Table 2. The pEV5HisTEV-*can1* wild-type and variant constructs were transformed into C43 (DE3) *E. coli* cells (Sigma Aldrich). Protein expression was induced with 0.4 mM isopropyl-β-D-1-thiogalactoside (IPTG) at an $OD_{600}$ of ~0.6 and grown overnight at 25 °C. Cells were harvested and resuspended in lysis buffer containing 50 mM Tris–HCl pH 7.5, 500 mM NaCl, 10 mM imidazole and 10% glycerol, and lysed by sonicating six times for 2 min on ice with 2 min rest intervals. Can1 was purified with a 5 ml HisTrapFF column (GE Healthcare), washed with 5 column volumes (CV) of buffer containing 50 mM Tris–HCl pH 7.5, 500 mM NaCl, 30 mM imidazole and 10% glycerol, and eluted with a linear gradient of buffer containing 50 mM Tris–HCl pH 7.5, 500 mM NaCl, 500 mM imidazole and 10% glycerol across 15 CV. Protein containing fractions were concentrated and the hexa-histidine affinity tag was removed by incubating protein with Tobacco Etch Virus (TEV) protease (10:1) overnight at room temperature.

Cleaved Can1 was isolated from the TEV protease by repeating the immobilised metal affinity chromatography step and collecting the unbound fraction. Size exclusion chromatography was used to further purify Can1, eluting protein isocratically with buffer containing 20 mM Tris-HCl pH 7.5, 150 mM NaCl. The protein was concentrated using a centrifugal concentrator, aliquoted and frozen at −80 °C. For seleno-methionine labelling, the pEV5HisTEV-Can1 construct was transformed into B834 (DE3) *E. coli* (Novagen) and cultures grown in M9 minimal medium supplemented with Selenomethionine Nutrient Mix (Molecular Dimensions, Newmarket, Suffolk, UK) and 50 mg $L^{−1}$ (L)-selenomethionine (Acros Organics). Protein was purified as described above.

**Can1 nuclease assay**. In total 200 nM Can1 was incubated with 2 nM plasmid pEV5HisTEV for the time indicated at 60 °C in 20 µl final reaction volume in 20 mM MES pH 6.5, 100 mM NaCl and 1 mM EDTA supplemented with 5 mM $MnCl_2$ and 200 nM $cA_4$. Reactions were stopped by addition of 10 mM EDTA. Control reactions were carried out by incubating plasmid without protein, $MnCl_2$ or $cA_4$ under the same conditions. For kinetic analysis, triplicate experiments were carried out by incubating Can1 in 140 µl final reaction volume under the conditions described above. 20 µl was removed and quenched by addition of 2 µl

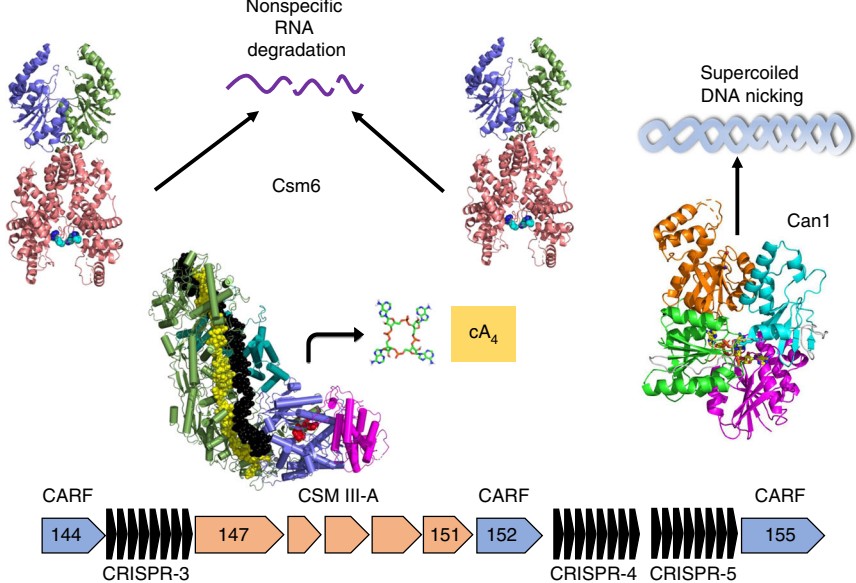

**Fig. 7 Cartoon of the type III-A CRISPR defence system of _T. thermophilus._** The type III-A effector binds phage RNA, activating the cyclase domain which synthesises cA$_4$ from ATP. cA$_4$ binds to and activates the two Csm6 family RNases (encoded by _tthb144_ and _tthb152_) and the DNA nickase Can1 (encoded by _tthb155_), providing a multi-layered defence against mobile genetic elements.

100 mM EDTA at desired time points. To differentiate nuclease activity of wild type Can1 and all variants, the kinetic assay was changed by reducing protein and cA$_4$ concentration to 100 nM and 20 nM, respectively. After adding 4 μl 6× DNA loading dye (New England BioLabs, Ipswich, MA, United States), 10 μl sample was analysed by 0.7% native agarose gel electrophoresis. Gels were scanned by Typhoon FLA 7000 imager (GE Healthcare) at a wavelength of 532 nm, quantified using the Bio-Formats plugin[43] of ImageJ as distributed in the Fiji package[44] and plotted against the time using Kaleidagraph (Synergy Software, Reading, PA, United States). The data were fitted to a single exponential curve[1].

**Nicking endonuclease digestion**. Nicking endonuclease Nt.BspQI (New England BioLabs, Ipswich, MA, United States) hydrolyses the phosphodiester bond between the 3'-hydroxyl and 5'-phosphate on one strand of dsDNA. Therefore, the open circle conformation of pEV5HisTEV containing the unique GCTCTTCN^ site was generated by Nt.BspQI. All incubations with commercial enzymes followed manufacturer's instructions.

**Alkaline agarose gel electrophoresis**. DNA products were separated on 0.7% alkaline agarose gels (30 mM NaCl, 2 mM EDTA and 0.7% agarose). The gels were immersed for 2 h in alkaline electrophoresis buffer (30 mM NaOH, 2 mM EDTA) before running. Samples and ladder were denatured with 6× alkaline electrophoresis loading dye (180 mM NaOH, 6 mM EDTA, 18% Ficoll 400 and 0.05% bromophenol blue) and heated at 95 °C for 5 min, and then chilled on ice. Gels were run at 2 V cm$^{-1}$ for 10 h in alkaline electrophoresis buffer, then soaked in renaturing buffer (500 mM Tris–HCl pH 8.0) for 30 min and stained with SYBR Gold (Thermo Scientific, Waltham, MA, United States) for 30 min. Gels were destained with H$_2$O and scanned by Typhoon FLA 7000 imager at a wavelength of 532 nm.

**Plasmid ligation and supercoiling**. In total 200 nM Can1 was incubated with 2 nM plasmid pEV5HisTEV in 20 μl final reaction volume as described above. Reactions were quenched and deproteinized by PCR Clean-Up System (Promega, Madison, Wisconsin, United States). In total 10 μl eluted product was incubated with T4 DNA Ligase (New England BioLabs, Ipswich, MA, United States) in ligase buffer (50 mM Tris–HCl pH 7.5, 10 mM MgCl$_2$, 1 mM ATP and 10 mM DTT) in 20 μl final reaction volume following manufacturer's instructions. After inactivating at 65 °C for 10 min, ligated products were incubated with _E. coli_ DNA gyrase (Inspiralis, Norwich, UK) in gyrase buffer (35 mM Tris–HCl pH 7.5, 24 mM KCl, 4 mM MgCl$_2$, 2 mM DTT, 1.8 mM spermidine, 1 mM ATP, 6.5% glycerol and 0.1 mg mL$^{-1}$ albumin) in 40 μl final reaction volume as per manufacturer's instructions. Reactions were deproteinized by phenol chloroform extraction before analysis by agarose electrophoresis.

**Preparation of cA$_4$**. cA$_4$ was purchased from Biolog Life Science Institute, Bremen, Germany. $^{32}$P labelled cA$_4$ was prepared by incubating 120 μg _S. solfataricus_ Csm complex with 0.5 mM ATP, 1 mM MgCl$_2$ and 100 nM target RNA in 20 mM MES pH 5.5, 100 mM K-glutamate and 1 mM DTT for 2 h at 70 °C. The reaction

product was isolated by phenol-chloroform extraction, followed by chloroform extraction, and frozen at −20 °C[25].

**Can1 and cA$_4$ electrophoretic mobility shift assay**. In total 100 nM $^{32}$P labelled cA$_4$ was incubated with different concentrations of Can1 for 10 min at room temperature in 10 μl final reaction volume in 20 mM MES pH 6.5, 100 mM NaCl, 1 mM EDTA, 5 mM MnCl$_2$ and 2 μM BSA. In total 10 μl of 20% glycerol was added to each reaction before loading onto native acrylamide gel (15% acrylamide, 1 × TBE). $^{32}$P labelled cA$_4$ was visualised by phosphorimaging using a Typhoon FLA 7000 imager (GE Healthcare). All images were analysed by Bio-Formats plugin[43] of ImageJ as distributed in the Fiji package[44].

**Crystallisation of Can1**. Selenomethionine labelled Can1 at 10.4 mg mL$^{-1}$ was mixed in a 1:2 molar ratio with cA$_4$ and incubated at room temperature for 30 min, before centrifugation at 13,000 rpm prior to crystallisation. Sitting drop vapour diffusion experiments were set up at the nanoliter scale using both commercially available and in-house crystallisation screens, and subsequently incubated at 293 K. Crystals appeared in many conditions, but those used for data collection grew from a reservoir solution of 20% PEG 3350, 200 mM sodium citrate, and 100 mM bis-Tris propane, pH 6.5. Crystals were harvested and transferred to a drop of reservoir solution with the addition of 20% glycerol before cryo-cooling. Attempts to crystallise Can1 in the absence of cA$_4$ did not yield crystals.

**X-ray data processing, structure solution, and refinement**. X-ray data were collected at a wavelength of 0.9159 Å, on beamline I04-1 at Diamond Light Source, at 100 K to 1.83 Å resolution. Data were automatically processed using Xia2[45] using XDS and XSCALE[46]. A strong anomalous signal was detected to 2.36 Å resolution. The data were phased using AutoSol in Phenix and the initial model was built in AutoBuild[47]. Model refinement was achieved by iterative cycles of REFMAC5[48] in the CCP4 suite[49] and manual manipulation in COOT[50]. Electron density for cA$_4$ was clearly visible in the maximum likelihood/$\sigma_A$ weighted $F_{obs}$—$F_{calc}$ electron density map at 3σ. The coordinates for cA$_4$ were generated in ChemDraw (Perkin Elmer) and the library was generated using acedrg[27], before fitting of the molecule in COOT. Model quality was monitored throughout using Molprobity[51] (score 1.05; centile 100). Ramachandran statistics are 98.5% favoured, 0% disallowed. Data and refinement statistics are shown in Supplementary Table 3. The coordinates and data have been deposited in the Protein Data Bank with deposition code 6SCE.

**Small angle X-ray scattering**. The apo-state of Can1 was measured using size exclusion chromatography coupled to small angle X-ray (SEC-SAXS) on beamline B21 at Diamond Light Source. A 4.8 ml Shodex KW-402.5 column was pre-equilibrated with 3 column volumes of buffer containing 20 mM Tris pH 7.5 and 150 mM NaCl. SEC-SAXS experiments were measured at a flow rate of 0.160 ml per minute with 2 s exposure per frame in a 1.5 mm diameter, 10 μm thick quartz capillary flow cell and a 800 μm squared X-ray beam focused on an Eiger 4 M detector 4 m from the sample. Column performance and instrumentation

calibration was checked by measuring a 45 µl sample of BSA injected at 10 mg mL$^{-1}$. In total 45 µl of the apo-state of Can1 was injected (10.4 mg mL$^{-1}$, 140 µM) and frames were collected for a total of 30 min. The flat radius-of-gyration, $R_g$, and nearly ideal overlay of the frames constituting the merged frames suggests the measured frames are free of interparticle interference (Supplementary Data 1).

To measure Can1 in the presence of cA$_4$, a SEC purification of Can1 was performed at the beamline by manually collecting the peak fraction (~100 µl) during an additional SEC run of the same 10.4 mg mL$^{-1}$ protein stock. The peak fraction represented a dilution of ~3.2× for a final concentration of 44 µM of purified protein. A corresponding buffer blank (~1 ml) was collected before sample injection. Samples were mixed with ligand to a final concentration of 0.25 mM and 3 dilutions of the sample were made first by making a two-thirds dilution of the master stock followed by two consecutive nine-tenths dilutions. Dilutions were performed with the buffer blank supplemented with ligand to maintain a ratio of 3-to-1 ligand-to-protein ratio. Samples were processed in batching mode at 25 µl per sample using the Arinax sample handling robot.

Raw SAXS images were processed with the DAWN[52] processing pipeline at the beamline to produce normalised, integrated 1-D un-subtracted SAXS curves. SEC-SAXS analysis and buffer subtractions were performed with the program ScÅtter (www.bioisis.net). Modelling of the apo-state was performed using the program CNS version 1.3 (cns-online.org/v1.3)[53]. First, missing loops and tails from the X-ray crystal structure were added by building an extended chain model of Can1 with generate_extended.inp. Protein only distance restraints were extracted from the ligand bound crystal structure of Can1 and used as pseudo-NOE constraints to fold the extended chain into the known parts of the crystal structure (model_anneal.inp). The resulting structure matched the crystal structure with missing loops and tails added back but in unconstrained positions. The folded structure (ground-state) was used as a starting conformation for limited rounds of simulated annealing or constant temperature simulations (anneal.inp) to sample various conformations as domain-to-domain distance restraints were varied. Specifically, the two CARF domains were kept as a rigid unit, whereas either the nuclease or nuclease-like domains were allowed to vary from their ground-state positions as rigid bodies. An initial simulation allowed the non-CARF domains to move as unconstrained rigid bodies only tethered by the linking peptide regions. Additional simulations were performed that constrained the non-CARF domains within 5 and 10 Å of their observed crystallographic positions. These were loose restraints added between the centre-of-masses for each domain based on the P(r)-distribution of the apo-state SAXS data. Specifically, no distances were allowed to exceed $d_{max}$ and distances were given upper and lower limits determined as $\pi/q_{max}$. A total of 2430 conformations were generated and used as an ensemble fit with the online web-app multi-FOXS[54] which identified the best two-state model for the apo-state. Single model fits were performed with FOXS (http://modbase.compbio.ucsf.edu/foxs)[55].

**Reporting summary**. Further information on research design is available in the Nature Research Reporting Summary linked to this article.

## Data availability

The final protein model and structure factors presented in this study have been deposited in the Protein DataBank with the accession code 6SCE. The SAXS data has been deposited in bioisis with the accession codes CAN1AP and CAN1C4. The source data underlying Figs. 4b–d, 5a, b and Supplementary Fig. 5a, b are provided as a Source Data file. Other data presented in this study are available from the corresponding authors upon reasonable request.

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

## Acknowledgements

We are grateful to the University of St Andrews Mass Spectrometry facility, and to staff at Diamond Light Source (beamlines I04-1 and B21). This work was supported by grants from the Biotechnology and Biological Sciences Research Council (REF: BB/S000313/1 to M.F.W. and REF: BB/R008035/1 to T.M.G.) and the China Scholarship Council (REF: 201703780015 to W.Z.).

## Author contributions

M.F.W., T.M.G., and R.R. planned the experiments. S.A.M. performed the crystallisation and X-ray crystallography work, W.Z. performed the biochemical assays, W.Z. and S.G. performed the gene cloning and protein expression, and R.R. performed the SAXS and MD studies. M.F.W., T.M.G., and R.R. analysed the data and wrote the paper, with input from all authors.

## Competing interests

The authors declare no competing interests.
