## [Peer Review File · Nature Communications]

Reviewers' Comments:

Reviewer #1:

Remarks to the Author:

Authors report activity and structure studies of a dual-CARF protein and its complex with the second messenger cA4. This work would be the first observation of a bound cA4 atomic structure if the work by Jia et al. was not published in Mol. Cell. However, the insights learned that how cA4 can interact with both CARF domains similarly as CARF dimers are still valuable and novel. In addition, the activity of this enzyme as a DNase activated by cA4 binding is also novel. Authors carefully characterized the nature of the DNase activity and the residues responsible for it. However, the most important question with regard to the molecular mechanism of Can1, that is, HOW cA4 activates DNA cleavage activity, remains partially answered. Even though authors clearly showed biochemically that binding of cA4 is necessary for the DNA cleavage activity, the molecular basis for activation is not clear due to the lack of an apo structure or that bound with DNA/cA4. This weakness can be partially addressed by performing additional mutagenesis experiment and perhaps more careful analysis of the structure.

Specific comments:

- 1) SAXS experiments provided an interesting observation that the enzyme becomes more compact upon cA4 binding. If authors believe that the SAXS data indicate a change in the nuclease domain conformation upon cA4 binding, can authors identify possible structural element that enables this conformational change such as the linker identified in MD simulation by performing mutagenesis analysis? Alternatively, Gly555 may facilitate this conformational change as well. So its disruption should impact cA4-activated DNA cleavage activity;
- 2) To show convincingly that cA4 activate DNase activity, authors should provide additional mutational data on residues that contact cA4 such as Trp42, His113, Asp13, Asn12, and Gln301. The activity of these mutants can also address the importance of the asymmetrical binding site;
- 3) Page 3, top, the description of the CARF fold here makes it sound like CARF has 7 beta strands (5 + a hairpin). Is it 7 or just 6 as typically found in CARF (where the hairpin is comprised of strands 5 and 6)? Perhaps authors can add a topology diagram with clear labels for each beta strand and alpha helix. This can make comparison of other CARF structures easier;
- 4) Figure 1, caption, the description says cA4 is shown in yellow spheres but it appears to be also represented by blue and pink spheres. The same description discrepancy is also found in Figure 2 caption;
- 5) Page 3, last paragraph. The description of the electron density for cA4 is confusing. It says Fo-Fc at 3 sigma while Figure 2A caption says 2Fo-Fc at 1.7 sigma. From the look of it, it appears to be a 2Fo-Fc map. However, it is better to identify bound ligands by Fo-Fc. It might be more convincing for authors to show Fo-Fc map (before cA4 was modeled in) in Figure 2A even if it may have lower quality because it would not have any model bias. Alternatively, authors can use the final refined model and compute an omit map either with 2Fo-Fc or 3Fo-2Fc as Fourier coefficients;
- 6) Authors refer to their structure as the "first" atomic structure of cA4. Since the paper by Jia et al., author should not claim the first. Authors should still describe detailed interactions because these appear to be quite different from those observed for Csm6-bound cA4;
- 7) Authors should label each of the four adenosine residues as A1, A2 etc. to differentiate them and to better describe their interactions;

- 8) Pages 3 & 4, last paragraph, it is incorrect to use the term "molecules" to describe adenosine residues in cA4;
- 9) Page 4, last paragraph, why do authors call the Trp42-stacked interaction "higher energy"? It should be a very favorable interaction, thus a lower energy;
- 10) Page 5, 3rd paragraph, there are two points here. First, authors discovered that Can1 has a structural homology to VC1899 and revealed nearly no conformational change in the dual CARF domains. If authors do not know if VC1899 is a functional cA4 binder, they should not suggest that cA4 binding does not cause a conformational change. Second, can authors compare the conformations of the two nuclease domains when the two protein's dual CARF domains are aligned (in Figure 3A, show nuclease domains)? In another word, are the nuclease domains placed similarly with respect to the dual CARF domains in the two structures? If not, this could suggest a conformational change in the nuclease domain in Can1 when cA4 is bound;
- 11) In several places throughout the manuscript, authors quoted RMSDs as over certain numbers of amino acids. It is likely they meant the mainchain atoms or the c-alpha atoms of certain number of amino acids, as structures being compared have different sequences;
- 12) Since we now know how Csm6 binds to cA4 (Jia et al.), it may be straightforward for authors to comment if Can1 uses similar types of residues to interact with cA4 and why it does not have ring-nuclease activity while Csm6 does;
- 13) Figure S7 caption does not make sense. Can1 nuclease domain is mentioned twice;
- 14)) It is surprising that anomalous signals were detected at 0.9159Å, an energy far removed from the narrow selenium edge (should be catered around 0.9792Å). Please check this inconsistency.
- 15) There are some grammatical errors throughout the manuscript that need to be revised. Additionally, there is mixed used of units such as "mM" and "M".

Reviewer #2:

Remarks to the Author:

The microbial CRISPR-Cas systems provide adaptive immunity against foreign nucleic acids, such as phages and plasmids. In the type III CRISPR-Cas systems, cyclic oligoadenylate (cOA) molecules act as a second messenger and activate the CARF family nucleases, such as the Csm6/Csx1 RNases and the Can1 DNases. In this study, McMahon et al. report the high-resolution crystal structure of Can1 bound to cA4, and revealed the detailed mechanism of cA4 recognition by Can1. Furthermore, a comparison of the Can1-cA4 structure with an apo-Can1 model predicted by SAXS, together with mutational analysis, provided mechanistic insights into cA4-mediated Can1 activation. Importantly, the high-resolution structure of the Can1-cA4 complex substantially improved our mechanistic understanding of cOA recognition by the CARF family proteins. The manuscript is well written, and the structure is nicely presented. Overall, I think that this manuscript will be suitable for publication in Nature Communications, if the authors address the following minor points.

1. P10. "As this is the first structure available for a CARF domain protein bound to its cognate cyclic oligoadenylate ligand, it is worth drawing some general conclusions about the mechanism of recognition." Since Jia et al. recently reported the crystal structure of Csm6 bound to cA4 (Jia et al.

Mol Cell 2019), it would be informative to include a comparison between Can1 and Csm6 in Discussion, regarding their overall structures, the bound cA4 conformations and cA4 recognition by their CARF domains.

2. Figure 1A. It may be better to change "Domain 2" to "Nuclease-like".
3. Figure 1B and 1C. It may be better to merge panels B and C and to add a label indicating the relationship between the two structures (e.g., an arrow indicating a rotation degree).
4. Figure 2A and 2B. It would be helpful to number the four AMP moieties of the bound cA4 molecule in the text and Figure 2.
5. Figure 6D. It would be informative to add labels indicating the states of the two structures (e.g., "Can1 (model)" and "Can1-cA4"). Also, the domains should be labeled.
6. Figure 6E. It would be better to show the structure in the same orientation as that of the complex structure in Figure 6D. Also, the domains should be labeled.
7. The authors should state how cA4 was prepared in Materials and Methods.

Reviewer #3:

Remarks to the Author:

I found the manuscript a pleasure to read. I believe that the authors have adequately reported the important information in regards to the SAXS data, and that the data is well analysed. I believe that the conclusions drawn from the SAXS data are well supported and not over-stated.

We would like to thank the reviewers for their comments, and address specific comments below.

Reviewer #1 (Remarks to the Author):

Authors report activity and structure studies of a dual-CARF protein and its complex with the second messenger cA4. This work would be the first observation of a bound cA4 atomic structure if the work by Jia et al. was not published in Mol. Cell. However, the insights learned that how cA4 can interact with both CARF domains similarly as CARF dimers are still valuable and novel. In addition, the activity of this enzyme as a DNase activated by cA4 binding is also novel. Authors carefully characterized the nature of the DNase activity and the residues responsible for it. However, the most important question with regard to the molecular mechanism of Can1, that is, HOW cA4 activates DNA cleavage activity, remains partially answered. Even though authors clearly showed biochemically that binding of cA4 is necessary for the DNA cleavage activity, the molecular basis for activation is not clear due to the lack of an apo structure or that bound with DNA/cA4. This weakness can be partially addressed by performing additional mutagenesis experiment and perhaps more careful analysis of the structure.

Specific comments:

1) SAXS experiments provided an interesting observation that the enzyme becomes more compact upon cA4 binding. If authors believe that the SAXS data indicate a change in the nuclease domain conformation upon cA4 binding, can authors identify possible structural element that enables this conformational change such as the linker identified in MD simulation by performing mutagenesis analysis? Alternatively, Gly555 may facilitate this conformational change as well. So its disruption should impact cA4-activated DNA cleavage activity;

We agree with the reviewer that it is important to consider the potential role of hinge regions in the reorganisation of the Can1 enzyme on cA4 binding. We have identified potential structural elements Gly423/Arg424/Pro425 and two PPG motifs that are suitably positioned to facilitate the conformational change. Given the residues we propose as potential hinges, as well as Gly550 suggested by the reviewer, are only able to interact via their main chain atoms, we felt that we would not be able to draw any conclusions about conformational change by mutating these residues to possess alternative side chains. Mutation of side chains such as Gly and Pro, which frequently have a structural role, would be unlikely to provide further clarity on the activation mechanism of Can1. We have therefore added these suggestions to the text as follows:

“A potential ‘hinge’ between the second CARF domain and the nuclease domain to allow these open and closed states is the connecting loop which contains a Gly423/Arg424/Pro425 motif. The loop between the first CARF domain and the nuclease-like domain is longer, and contains two ‘PPG’ motifs (Pro160/Pro161/Gly162 and Pro165/Pro166/Gly167) which are likely to be important in providing flexibility between the domains.”

2) To show convincingly that cA4 activate DNase activity, authors should provide additional mutational data on residues that contact cA4 such as Trp42, His113, Asp13, Asn12, and Gln301. The activity of these mutants can also address the importance of the asymmetrical binding site;

To address this point, we have expanded our analysis of the cA₄ binding site significantly by constructing and analysing four new enzyme variants: N12A, W42A, H113A and Q222E. We

provide new data on cA₄ binding and DNA cleavage for the wild-type and variant proteins (new Supplementary Figure 5), along with accompanying text. We demonstrate that the largest effect on both cA₄ binding and catalysis is observed in the K90E variant. The other variants display varied reductions in cA₄ binding affinity and modest reductions in nuclease activity. These new data enhance our understanding of the cA₄ binding site of Can1 and its relation to DNA nuclease activity. The other residues mentioned by the reviewer interact with cA₄ by hydrogen bonds with main chain atoms only, and so we felt we could not draw conclusions from mutating the side chains for these.

3) Page 3, top, the description of the CARF fold here makes it sound like CARF has 7 beta strands (5 + a hairpin). Is it 7 or just 6 as typically found in CARF (where the hairpin is comprised of strands 5 and 6)? Perhaps authors can add a topology diagram with clear labels for each beta strand and alpha helix. This can make comparison of other CARF structures easier;

The CARF domains in Can1 each have 7 beta-strands. The presence of an additional beta-strand when compared to other reported CARF domain structures has now been noted in the text. In addition, a topology figure has been added as a supplementary information (Figure S1).

4) Figure 1, caption, the description says cA₄ is shown in yellow spheres but it appears to be also represented by blue and pink spheres. The same description discrepancy is also found in Figure 2 caption;

For structural figures, it is conventional to mention the colour of the carbon atoms, and the blue for nitrogen and red for oxygen is usually inferred. This has now been explicitly described in the Figure 1 and 2 legends.

5) Page 3, last paragraph. The description of the electron density for cA₄ is confusing. It says Fo-Fc at 3 sigma while Figure 2A caption says 2Fo-Fc at 1.7 sigma. From the look of it, it appears to be a 2Fo-Fc map. However, it is better to identify bound ligands by Fo-Fc. It might be more convincing for authors to show Fo-Fc map (before cA₄ was modeled in) in Figure 2A even if it may have lower quality because it would not have any model bias. Alternatively, authors can use the final refined model and compute an omit map either with 2Fo-Fc or 3Fo-2Fc as Fourier coefficients;

Figure 2 has been changed to show the Fo-Fc map (at 3 sigma) to ensure consistency with the text.

6) Authors refer to their structure as the “first” atomic structure of cA₄. Since the paper by Jia et al., author should not claim the first. Authors should still describe detailed interactions because these appear to be quite different from those observed for Csm6-bound cA₄;

References to this being the first structure in complex with cA₄ have been removed.

7) Authors should label each of the four adenosine residues as A1, A2 etc. to differentiate them and to better describe their interactions;

This has been done, with labels added both in the relevant figures and text.

8) Pages 3 & 4, last paragraph, it is incorrect to use the term “molecules” to describe adenosine residues in cA₄;

In these cases ‘molecules’ has been changed to ‘moieties’.

9) Page 4, last paragraph, why do authors call the Trp42-stacked interaction “higher energy”? It should be a very favorable interaction, thus a lower energy;

The higher energy was referring to the adenine base rather than Trp42, but it was ambiguous in the text. We have now made this sentence clearer.

10) Page 5, 3rd paragraph, there are two points here. First, authors discovered that Can1 has a structural homology to VC1899 and revealed nearly no conformational change in the dual CARF domains. If authors do not know if VC1899 is a functional cA4 binder, they should not suggest that cA4 binding does not cause a conformational change. Second, can authors compare the conformations of the two nuclease domains when the two protein's dual CARF domains are aligned (in Figure 3A, show nuclease domains)? In another word, are the nuclease domains placed similarly with respect to the dual CARF domains in the two structures? If not, this could suggest a conformational change in the nuclease domain in Can1 when cA4 is bound;

For the first point, we have modified the text to introduce the caveat that we do not know for certain that VC1899 binds cA4. In the “*Structural comparisons of the cA₄ binding site*”, we have added the final sentence has been altered to “In other words, with the caveat that VC1899 has not yet been demonstrated to bind cA₄, the cA₄ binding site of Can1 is likely to be largely pre-formed in the apo-protein.”

For the second point, we have included the figure (Supplementary Figure 10) as suggested the reviewer, and added some text “*A superimposition of Can1 onto VC1899, structurally aligned on the CARF domain dimer from each protein only, suggests the possibility of movement of the nuclease and nuclease-like domains between the open and closed states (Supplementary Figure 10). It is possible that a complex population of open and closed structures exists in the absence of cA₄.*”

11) In several places throughout the manuscript, authors quoted RMSDs as over certain numbers of amino acids. It is likely they meant the mainchain atoms or the c-alpha atoms of certain number of amino acids, as structures being compared have different sequences;

We have qualified this by changing ‘residues’ to ‘C α atoms’ where RMSD values are quoted.

12) Since we now know how Csm6 binds to cA4 (Jia et al.), it may be straightforward for authors to comment if Can1 uses similar types of residues to interact with cA4 and why it does not have ring-nuclease activity while Csm6 does;

We agree with this point and have now added a paragraph comparing the cA4 interaction interfaces of Csm6, Csx1 and Can1: “*Recently, the structures of two CARF-HEPN family ribonucleases, Csm6 from *Thermococcus onnurineus* and Csx1 from *S. islandicus* have been solved with cA₄ bound^{31,32}. Although each of these proteins have the basic unit of a CARF domain for cA₄ binding in common with Can1, there is little sequence conservation across the three proteins. In particular, Can1 lacks conserved Y14, Y19, S51, H155 and N158 cA₄-interacting residues identified as conserved across the Csx1/Csm6 family³². N158 is replaced by K90 in Can1, reflecting the importance of this position in the centre of the CARF domain for cA₄ binding despite a lack of sequence conservation. Residues W14 and H132, identified as important for cA₄ cleavage by Csm6³¹, are also not conserved in Can1, and we see no evidence for ring nuclease activity by Can1 in vitro. The three enzymes bind cA₄ in different*

orientations, emphasising the conformational flexibility possible for this large cyclic nucleotide.”

13) Figure S7 caption does not make sense. Can1 nuclease domain is mentioned twice;

The first mention of Can1 is the title of the figure, which is shown in bold, and the second time refers to the detail of what each panel shows. The same pattern is used for all figures throughout the paper and follows the *Nature Communications* formatting guidelines.

14)) It is surprising that anomalous signals were detected at 0.9159A, an energy far removed from the narrow selenium edge (should be catered around 0.9792A). Please check this inconsistency.

The wavelength stated in the Methods is correct. Diamond Light Source beamline I04-1 is not tunable, but claims that Se SAD is possible at this wavelength if the data are collected carefully (<https://www.diamond.ac.uk/Instruments/Mx/I04-1.html>). The anomalous difference at 0.92 Å is still significant and sufficient to phase the data. We have now solved 2 structures using Se SAD on this beamline.

15) There are some grammatical errors throughout the manuscript that need to be revised. Additionally, there is mixed used of units such as “mM” and “M”.

All units have been converted into mM for consistency. We have corrected all grammatical errors we could identify, but if further specific examples could be provided we would be happy to address them.

Reviewer #2 (Remarks to the Author):

The microbial CRISPR-Cas systems provide adaptive immunity against foreign nucleic acids, such as phages and plasmids. In the type III CRISPR-Cas systems, cyclic oligoadenylate (cOA) molecules act as a second messenger and activate the CARF family nucleases, such as the Csm6/Csx1 RNases and the Can1 DNases. In this study, McMahon et al. report the high-resolution crystal structure of Can1 bound to cA4, and revealed the detailed mechanism of cA4 recognition by Can1. Furthermore, a comparison of the Can1-cA4 structure with an apo-Can1 model predicted by SAXS, together with mutational analysis, provided mechanistic insights into cA4-mediated Can1 activation. Importantly, the high-resolution structure of the Can1-cA4 complex substantially improved our mechanistic understanding of cOA recognition by the CARF family proteins. The manuscript is well written, and the structure is nicely presented. Overall, I think that this manuscript will be suitable for publication in *Nature Communications*, if the authors address the following minor points.

1. P10. “As this is the first structure available for a CARF domain protein bound to its cognate cyclic oligoadenylate ligand, it is worth drawing some general conclusions about the mechanism of recognition.” Since Jia et al. recently reported the crystal structure of Csm6 bound to cA4 (Jia et al. *Mol Cell* 2019), it would be informative to include a comparison between Can1 and Csm6 in Discussion, regarding their overall structures, the bound cA4 conformations and cA4 recognition by their CARF domains.

We agree with this point and have now added a paragraph comparing the cA4 interaction interfaces of Csm6, Csx1 and Can1: “Recently, the structures of two CARF-HEPN family ribonucleases, Csm6 from *Thermococcus onnurineus* and Csx1 from *S. islandicus* have been solved with cA₄ bound^{31,32}. Although each of these proteins have the basic unit of a CARF

domain for cA₄ binding in common with Can1, there is little sequence conservation across the three proteins. In particular, Can1 lacks conserved Y14, Y19, S51, H155 and N158 cA₄-interacting residues identified as conserved across the Csx1/Csm6 family³². N158 is replaced by K90 in Can1, reflecting the importance of this position in the centre of the CARF domain for cA₄ binding despite a lack of sequence conservation. Residues W14 and H132, identified as important for cA₄ cleavage by Csm6³¹, are also not conserved in Can1, and we see no evidence for ring nuclease activity by Can1 in vitro. The three enzymes bind cA₄ in different orientations, emphasising the conformational flexibility possible for this large cyclic nucleotide.”

2. Figure 1A. It may be better to change “Domain 2” to “Nuclease-like”.

We have now included both names in the legend – at this point in the article the term ‘nuclease-like domain’ has not been introduced.

3. Figure 1B and 1C. It may be better to merge panels B and C and to add a label indicating the relationship between the two structures (e.g., an arrow indicating a rotation degree).

We have now done this, thanks for the suggestion.

4. Figure 2A and 2B. It would be helpful to number the four AMP moieties of the bound cA₄ molecule in the text and Figure 2.

As suggested, we have now named the AMP moieties A1-A4 in Figure 2, other relevant supplementary figures, and the text.

5. Figure 6D. It would be informative to add labels indicating the states of the two structures (e.g., “Can1 (model)” and “Can1-cA₄”). Also, the domains should be labeled.

These labels have been added.

6. Figure 6E. It would be better to show the structure in the same orientation as that of the complex structure in Figure 6D. Also, the domains should be labeled.

The orientation has been changed to match that in Figure 6D, and the domains have been labelled.

7. The authors should state how cA₄ was prepared in Materials and Methods.

This information has been added to the Materials and Methods section.

Reviewer #3 (Remarks to the Author):

I found the manuscript a pleasure to read. I believe that the authors have adequately reported the important information in regards to the SAXS data, and that the data is well analysed. I believe that the conclusions drawn from the SAXS data are well supported and not over-stated.

We appreciate the supportive feedback from this reviewer, who recognises the key contribution of the SAXS data in the inference of a structural reorganisation by Can1 on cA₄ binding.

Reviewers' Comments:

Reviewer #1:

Remarks to the Author:

Authors revised the manuscript to satisfaction and it is now ready to be published.